# Unravelling the effect of charge dynamics at the plasmonic metal/semiconductor interface for $CO_2$ photoreduction

Laura Collado[1,2], Anna Reynal[3,4], Fernando Fresno [1], Mariam Barawi[1], Carlos Escudero[5], Virginia Perez-Dieste[5], Juan M. Coronado [2], David P. Serrano[2,6], James R. Durrant[3] & Víctor A. de la Peña O'Shea [1]

Sunlight plays a critical role in the development of emerging sustainable energy conversion and storage technologies. Light-induced $CO_2$ reduction by artificial photosynthesis is one of the cornerstones to produce renewable fuels and environmentally friendly chemicals. Interface interactions between plasmonic metal nanoparticles and semiconductors exhibit improved photoactivities under a wide range of the solar spectrum. However, the photo-induced charge transfer processes and their influence on photocatalysis with these materials are still under debate, mainly due to the complexity of the involved routes occurring at different timescales. Here, we use a combination of advanced in situ and time-resolved spectroscopies covering different timescales, combined with theoretical calculations, to unravel the overall mechanism of photocatalytic $CO_2$ reduction by $Ag/TiO_2$ catalysts. Our findings provide evidence of the key factors determining the enhancement of photoactivity under ultraviolet and visible irradiation, which have important implications for the design of solar energy conversion materials.

[1] Photoactivated Processes Unit, Institute IMDEA Energy, Avda. Ramón de la Sagra 3, 28935 Madrid, Spain. [2] Thermochemical Processes Unit, Institute IMDEA Energy, Avda. Ramón de la Sagra 3, 28935 Madrid, Spain. [3] Department of Chemistry and Centre for Plastic Electronics, Imperial College London, Exhibition Road, London SW7 2AZ, United Kingdom. [4] School of Science, Engineering and Design, Teesside University, Borough Road, Middlesbrough TS1 3BA, United Kingdom. [5] ALBA Synchrotron Light Source, Carretera BP 1413 Km. 3.3, Cerdanyola del Vallès, Spain. [6] Department of Chemical and Energy Technology, Rey Juan Carlos University, c/ Tulipán s/n, 28935 Madrid, Spain. These authors contributed equally: Laura Collado, Anna Reynal. Correspondence and requests for materials should be addressed to D.P.S. (email: david.serrrano@imdea.org) or to J.R.D. (email: j.durrant@imperial.ac.uk) or to V.Peña.O'Shea. (email: victor.delapenya@imdea.org)

Research efforts to develop sustainable and efficient carbon-neutral energy technologies are crucial to limit the harmful effects of greenhouse gas emissions associated with the fulfilment of the future energy demand[1,2]. To reach this goal, it is essential to achieve the improvement and deployment of $CO_2$ capture and utilisation technologies[3,4]. One of the most significant scientific challenges today is the harnessing of sunlight to obtain fuels and chemicals by artificial photosynthesis[5–7]. To succeed in this task and improve the current quantum yields, it is necessary to focus on the design of efficient photocatalysts capable of successfully managing photons and electrons, boosting the light absorption across the entire solar spectrum and enhancing charge separation and transport processes[8–10].

An extensive number of heterogeneous $CO_2$ reduction photocatalysts have been developed in the last years[11–13]. Amongst them, $TiO_2$ remains the most widely used, accounting for more than 50% of the published literature in this field[12]. Nevertheless, it possesses a series of limitations, such as a relatively poor conductivity, high overpotential for cathodic reactions, high recombination of the photo-induced charges and an ultraviolet (UV)-matching band gap, which only enables the use of a small fraction (~4%) of the solar spectrum. To overcome these limitations, different strategies have been developed, including band structure engineering, doping, sensitising, the use of co-catalysts and the formation of heterojunctions with other semiconductors[9,14,15].

In this way, plasmon-induced photocatalytic processes have gained attention due to their photoactivation tunability, extended optical absorption in a wide range of the visible spectrum, enhanced charge separation lifetimes, improved conversions and controlled selectivity when metal nanoparticles (NPs) act as co-catalysts[16–19].

Whilst the physics of the localised surface plasmon effect are well understood, several routes have been proposed to explain the influence of interfacial charge dynamics in photocatalytic reactions driven by metal NP/semiconductor systems: hot electron injection, resonance energy transfer, plasmon-exciton coupling, field nanoconfinement and direct electron injection[20–23]. However, an unequivocal explanation, especially on non-model, or real, catalysts, is still unclear and requires a holistic understanding of each and every process comprised in the overall reaction.

Herein, we combine unique in situ techniques and time-resolved optoelectronic characterisation with theoretical calculations to determine the aspects not fully understood about the photo-induced interfacial charge dynamics and the role of intermediates and oxygen in the $CO_2$ photoreduction reaction mechanism over Ag/$TiO_2$ photocatalysts, using $H_2O$ as electron donor and both UV and visible-light irradiation. Our study reveals that the plasmonic metal/semiconductor junction can introduce synergetic optical and electronic effects that modify the interfacial charge generation and electron transfer kinetics, driving $CO_2$ photoreduction towards highly electron-demanding products. The comprehension of charge photogeneration and transfer kinetics and mechanisms at the interface between plasmonic metals and semiconductors has a multidisciplinary impact because it is a powerful tool to design and optimise multi-electron transfer reactions, not only for artificial photosynthesis but also for new light-induced catalytic reactions.

## Results

**Photocatalyic activity.** UV-driven $CO_2$ photoreduction experiments show that homogenously distributed Ag NPs on the anatase $TiO_2$ surface (Fig. 1 and Supplementary Figures 1 and 2a–o) lead to a nearly 15-fold enhancement in the $CH_4$ production rate compared to the bare semiconductor (Fig. 1a), with which CO is obtained as the main product. On the contrary, under visible irradiation $TiO_2$ is highly selective to methanol (Fig. 1b). This behaviour is observed in all samples containing different Ag loadings (Fig. 1c and Supplementary Table 2). It should be noted that the reported $CH_4$ production rate (ca. 5.8 µmolg$_{cat}^{-1}$ h$^{-1}$) for 1.5Ag/$TiO_2$ catalyst (1.5 wt% Ag) outperforms most of the reported state-of-the-art values with Ag-based photocatalysts (Supplementary Table 3), using $H_2O$ as electron donor under mild operation conditions. Further increase in the silver loading leads to a lower hydrocarbon production that could be ascribed to different factors such as a lower Ag dispersion and surface area (Supplementary Table 1 and Supplementary Figure 2o), or a shading effect that hinders the light absorption by the semiconductor. Previous studies performed in our group attributed the change in selectivity to the electron scavenging ability of metal NPs, which retards the recombination and favours the formation of highly electron-demanding products (such as $CH_4$)[16,24]. However, the role of Ag NPs in charge dynamics and their effect on the photocatalytic behaviour are still unclear. To shed light on this point it is necessary to understand the photoelectron dynamics pathways at the Ag/$TiO_2$ interface and their effects on the catalytic mechanism (Fig. 1d).

**Electronic band structure of the Ag/$TiO_2$ heterojunction.** The combination of the valence band (VB) analysis from X-ray photoelectron spectroscopy (XPS), electrochemical impedance spectroscopy (EIS) and theoretical calculations allow to draw a complete scheme of the electronic properties of the metal/semiconductor interface (Fig. 2). This information is essential to understand the light-induced processes in photocatalytic reactions. Silver deposition on the titania surface modifies the interface band structure as confirmed by the formation of surface states near the $TiO_2$ Fermi level ($E_F$). This can be observed in the VB XPS spectrum (Fig. 2a) and also in the flat band potentials determined by the Mott–Schottky plots obtained through EIS measurements (Fig. 2c)[25]. The VB spectrum of bare titania shows a broad band due to O 2p that is formed by the characteristic contribution of bonding (at 7.6 eV) and non-bonding orbitals (at 5.6 eV). On the other hand, the Ag/$TiO_2$ sample shows an increase in surface states in the non-bonding region and changes in the VB edge (≈0.2 eV), which are consistent with an interface interaction between surface atoms that could be assigned to the formation of Ag–O bonds. This hypothesis is confirmed by the electronic structure obtained by density functional theory (DFT) calculation of Ag clusters supported on $TiO_2$ NPs. Thus, the density of states of a bare $TiO_2$ cluster shows a band gap ($E_g = 3.2$ eV) in which valence and conduction bands are mainly formed by O 2p and Ti 3d orbitals, respectively (Supplementary Figure 3). Otherwise, Ag/$TiO_2$ (Fig. 2b) exhibits the formation of induced interface states (IFS)[23] in the band gap region. These surface sub-band gap ($S_{Eg}$) states can be assigned to charge donation from Ag 5s to O 2p neighbouring atoms and Ti 3d orbitals. These $S_{Eg}$ states have been previously described in this kind of materials[26] and are consistent with ultrafast (<10 fs) photo-induced hot electrons in $TiO_2$, rather than electrons photogenerated in Ag and transferred to the semiconductor. The formation of these IFS states could be a cornerstone to explain photocatalytic activity in the visible spectrum region.

Because of the narrow differences between the Ag and $TiO_2$ work functions, depending on morphological or structural properties[27], different band-bending scenarios have been previously described for Ag/$TiO_2$ interfaces[22,24]. For the present samples, the flat band potentials and therefore the Fermi level ($E_F$) were calculated from the charge transfer capacitance ($C_{SC}$) obtained from the EIS measurements (Fig. 2c). The dependence of $C_{SC}$ on bias potential ($V$) is described by the Mott–Schottky

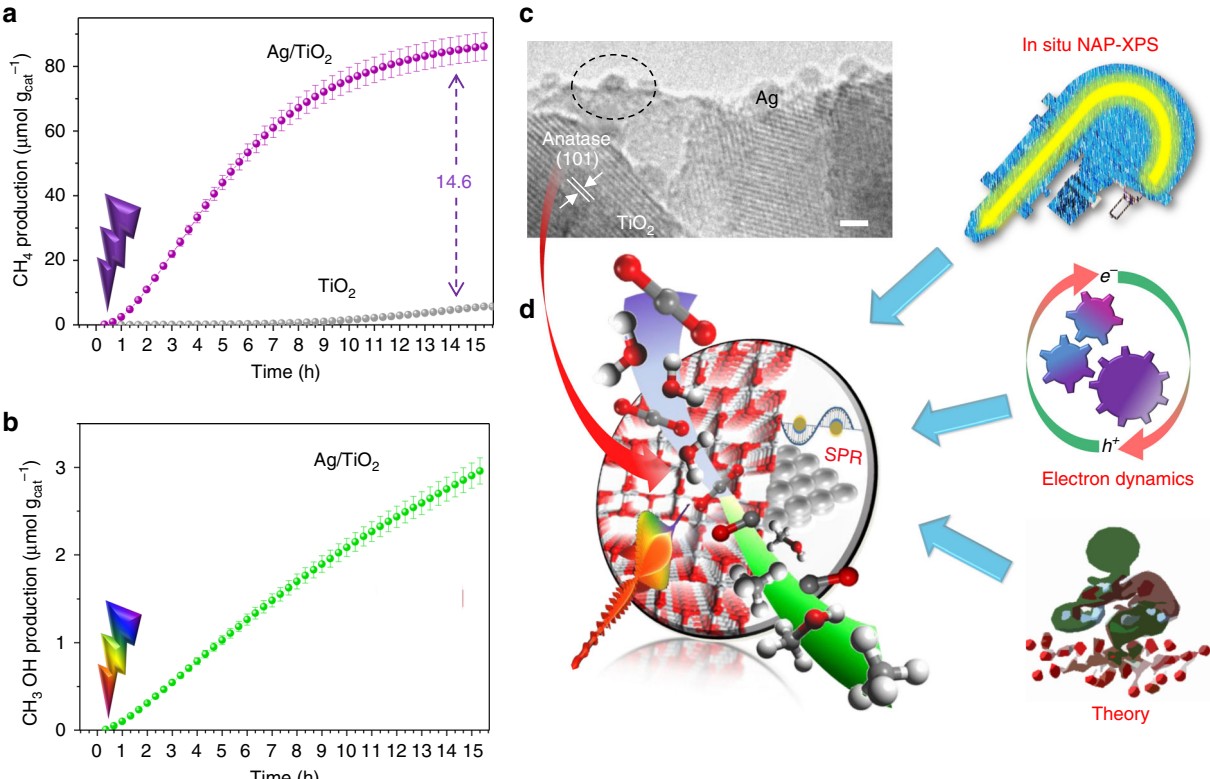

**Fig. 1** Light-induced charge dynamics pathways and their effect on $CO_2$ photoreduction. Cumulative production of **a** methane and **b** methanol under 365 nm and 450 nm irradiation, respectively. **c** Transmission electron microscopy (TEM) image of Ag nanocluster supported on the anatase $TiO_2$ surface, scale bar 2 nm. **d** A combination of in situ characterisation tools at different timescales and theoretical calculations allow the determination of the optoelectronic and surface properties at the Ag/$TiO_2$ interface and their effect on photoredox reactions in artificial photosynthesis

equation (Supplementary Note 1). Ag/$TiO_2$ presents a slightly positive shift on the flat band potential value ($\phi_B = 0.10 \pm 0.01$ eV) compared with $TiO_2$, which reveals a variation in the Fermi level energy when silver is deposited on the surface. This result indicates an interfacial charge transfer from $TiO_2$ to Ag that causes an upward band bending that is screened over a depletion layer thickness ($L_D$) that depends on the Ag particle size (Supplementary Figure 4). In addition, the carriers density ($N_d$) for Ag/$TiO_2 = =3.20 \times 10^{17}$ cm$^{-3}$ increased compared to bare $TiO_2 = 2.69 \times 10^{16}$ cm$^{-3}$, as determined from the slide of the Mott–Schottky plot. In addition, electron localisation function plots (Fig. 2d) show an evident polarisation in Ag atoms near the O sites, confirming the formation of hybridised orbitals due to the presence of Ag–O bonds.

Up to here, surface VB studies, electrochemical measurements and DFT calculations altogether allow to describe the Ag/$TiO_2$ interfaces as complex electronic structures resulting from the formation of IFS states due to the charge donation between surface Ti–O and Ag neighbours. In addition, this interaction also leads to changes in carrier density and in the band bending.

**Mechanistic studies of $CO_2$ photoreduction.** In situ near-ambient pressure XPS (NAP-XPS) under UV irradiation reveals a first step of $CO_2$ adsorption/activation, via carbonate/bicarbonate formation, followed by the preferential development of hydrocarbon intermediates. In ultra-high vacuum (UHV), the C1s signal of a fresh Ag/$TiO_2$ sample (Fig. 3a) shows the presence of spurious carbon (C–C, 284.7 eV), as well as other contributions assigned to bicarbonate ($HCO_3^-$, 288.6 eV), carbonate ($CO_3^{2-}$, 289.5 eV) and other species containing C–O bonds (285.5 eV) [26,28,29]. In turn, the O1s region (Fig. 3b) reveals the presence of

bridge oxygen species ($O_B$, 530.1 eV), hydroxyl groups ($O_{OH}$, 531.1 eV), different inorganic carbon–oxygen species (C–O*, 532.0 eV) and physisorbed water ($H_2O_{phys}$, 532.5 eV) [30,31]. After exposing the sample to a $CO_2$ and $H_2O$ atmosphere an increase of carbonate/bicarbonate species is observed as a result of the $CO_2$/$H_2O$ equilibrium on the catalyst surface (detailed information in Supplementary Tables 5 and 6) [28,32].

The irradiation of the sample with UV light leads to substantial changes. The most relevant finding in the C 1s spectrum is the appearance of two new components at 284.0 and 287.0 eV, corresponding to methylene and formate intermediates, respectively (Fig. 3a) [33,34]. In addition, two other components are observed at 291.9 and 292.9 eV associated with $CO_2^{\delta-}$ and desorbed gas-phase $CO_2$ [35], respectively. Moreover, a broadening occurs in the peaks of $HCO_3^-$ and $CO_3^{2-}$, which could also act as hole scavengers [36], while the peak at 287.01 eV can be attributed to the contribution of carbonyl intermediates [28,37]. During the reaction a decrease in OH groups and physisorbed $H_2O$ is observed in the O1s spectrum (Fig. 3b), indicating the role that both species are playing in the photoredox process. In summary, NAP-XPS observations in Ag/$TiO_2$ reveals an initial $CO_2$ adsorption/activation step via carbonate/bicarbonate species followed by the formation of methylene and formate species, which are tentatively identified as the hydrocarbons intermediates through a carbene reaction pathway [38]. In addition, Ag 3d signals in UHV and dark conditions reveal the coexistence of $Ag^0$ and $Ag^+$ species (Supplementary Figure 6A). After UV irradiation, the $Ag^0$ content increases due to the progressive reduction of silver oxides (Supplementary Figure 6B).

Raman studies (Supplementary Figure 7) show that the fresh Ag/$TiO_2$ surface is covered by different carbon species adsorbed

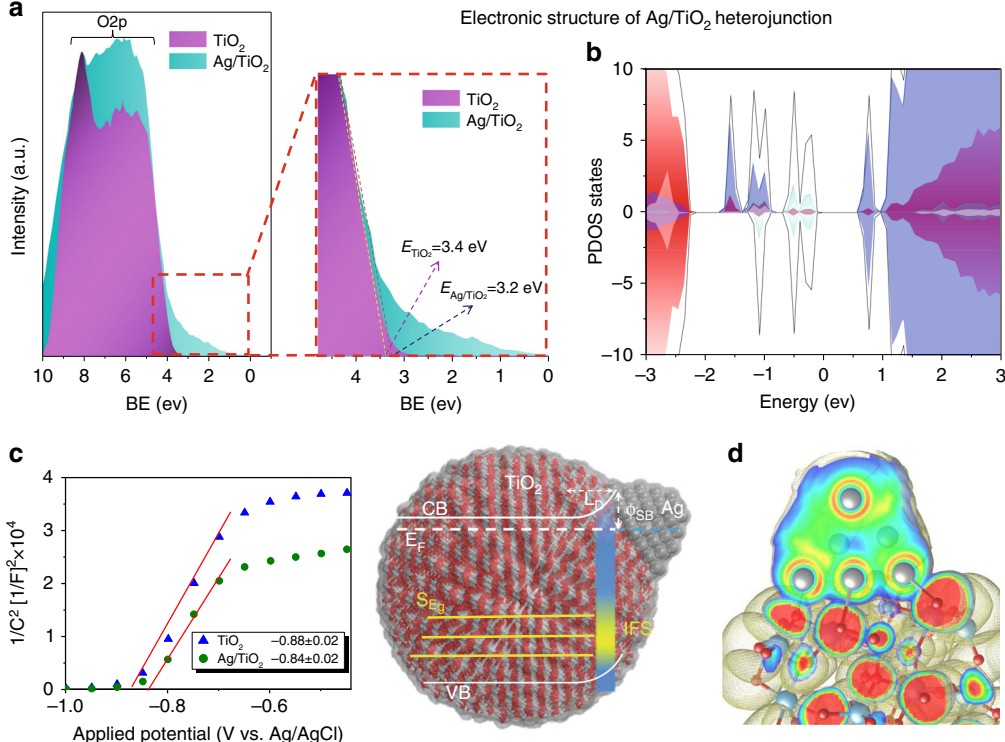

**Fig. 2** Scheme of electronic structure of Ag/TiO$_2$ interfaces. **a** Valence band XPS spectra of TiO$_2$ and 1.5Ag/TiO$_2$. **b** Total DOS (black) and atom-projected (PDOS) for Ag 5s (light blue), Ag 5d (grey) and Ti 3d (purple). **c** Mott–Schottky plots of TiO$_2$ and Ag/TiO$_2$ samples obtained with semiconductor-electrolyte interface capacity values acquired by EIS at 400 Hz at different bias potentials. **d** ELF isosurfaces and sections for Ag/TiO$_2$ cluster; atom colour: Ag (grey), Ti (green) O (red); BE binding energy, CB conduction band, VB valence band S$_{Eg}$ sub-band gap states, IFS induce interface states, $\phi_B$ bending energy, $L_D$ depletion layer thickness

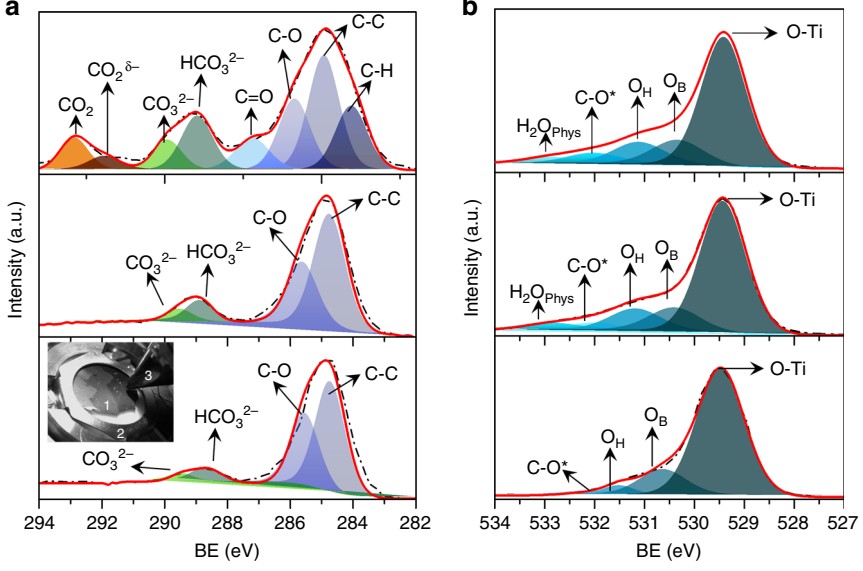

**Fig. 3** In situ NAP-XPS spectra of C1s **a** and O1s **b** signals recorded for Ag/TiO$_2$ under the following conditions (from bottom to top): UHV; after dosing CO$_2$ and H$_2$O ($P_{CO2} = 4.375 \times 10^{-2}$ mbar and $P_{H2O}$ 6.250 × 10$^{-3}$ mbar; $P_{Total} = 5 \times 10^{-2}$ mbar); and subsequently exposed to UV irradiation (365 nm)

from ambient air. The observed bands are assigned to linear and bent adsorbed CO$_2$, bicarbonate as well as bidentate and monodentate carbonates[39]. After reaction, some changes are observed, mainly related to a significant increase of adsorbed CO$_3^{2-}$ and HCO$_3^-$ species and the appearance of formate species (1366 cm$^{-1}$) as a possible reaction intermediate[40], in good agreement with NAP-XPS studies. The increase in the bands

due to bent CO$_2$ structures may be the consequence of the generation of CO$_2^-$ species coordinated to metal ions. In addition, it is worth highlighting the appearance of a new Raman band at a shift of ca. 910 and 985 cm$^{-1}$ assigned to the $\nu_{O-O}$ vibrational mode of surface peroxo Ti(O$_2$)[41] and peroxocarbonate species (CO$_4^{2-}$ and C$_2$O$_6^{2-}$)[39,42], respectively. This observation suggests that the formation of both peroxidised species arise, on

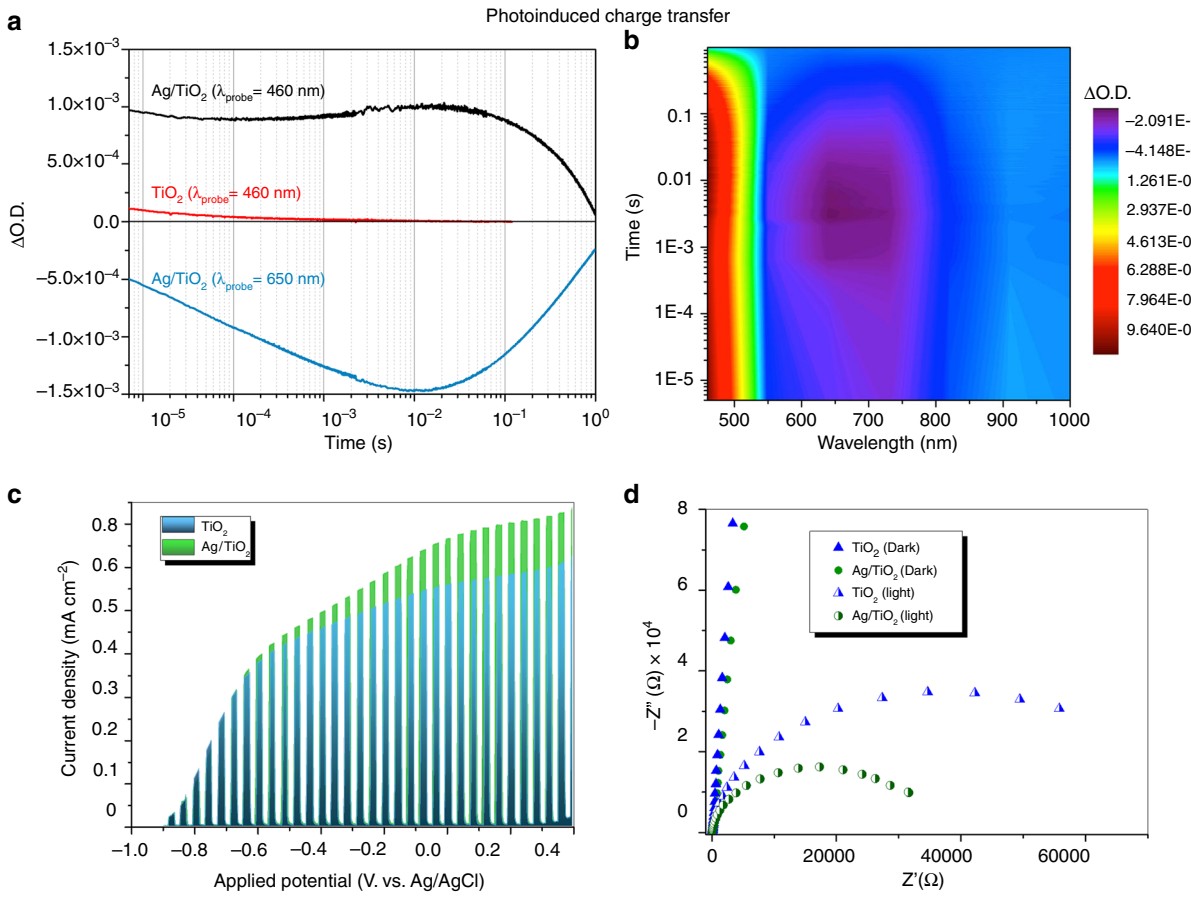

**Fig. 4** Photo-induced charge transfer in the Ag–TiO₂ interface: **a** Temporal profiles of transient absorption decays of Ag/TiO₂ and TiO₂ films, probed at 460 and 650 nm after excitation at 355 nm (350μJcm⁻², 1 Hz) and **b** two-dimensional plot of the transient absorption spectrum of an Ag/TiO₂ (5 s) film after UV excitation (355 nm, 350μJcm⁻²). **c** Photocurrents at different applied potentials under UV irradiation and **d** electrochemical impedance spectroscopy at 0 V vs. Ag/AgCl for TiO₂ and Ag/TiO₂ under dark and illumination conditions (inset: electrical circuit used for the data fitting)

the one hand, from hole capture by $CO_3^{2-}$ and $HCO_3^-$ and, on the other hand, from partial oxidation of water. This could prevent $O_2$ from being detected as the outcome of $H_2O$ oxidation.

**Photo-induced charge transfer.** In situ charge transfer studies using UV–vis (UV–visible) diffuse reflectance spectroscopy (DRS) at 365 nm irradiation (Supplementary Figure 8) show the quick formation of a broad band assigned to the surface plasmon resonance (SPR) of silver[43,44]. Before irradiation, only an intense absorption band with an onset at 380 nm is observed due to the band gap transition of TiO₂ (c.a. 3.2 eV). After 1 second of UV irradiation, a visible absorption appears as a broad band covering the range 400 - 700 nm and centred at about 500 nm, which is attributed to the SPR absorption of Ag⁰ NPs[43,44]. During UV irradiation, the intensity of the plasmon band increases with a red shift in the peak position (~80 nm), indicating a slight increase in the Ag particle size[22]. It is also noteworthy that after 5 min the plasmonic band did not show any significant change, indicating the high rate of Ag NPs photoreduction (Supplementary Figure 8). A concomitant decrease in the intensity of the UV absorption band is also observed. This behaviour is representative of the progressive formation of Ag⁰ due to the flow of photo-generated electrons in the conduction band of TiO₂ towards surface Ag oxides. The formation of Ag⁰ species is consistent with previous NAP-XPS observations (Supplementary Figure 6), and it is clearly noticeable by a darkening of the sample due to Ag NPs photochromism (Fig. 3a inset)[45].

Elucidation of photo-induced charge transfer dynamics is vital to understand the subsequent photoredox reactions, and therefore they are addressed herein by the use time-resolved spectroscopy and photoelectrochemical characterisation techniques (Fig. 4). Transient absorption spectroscopy (TAS) experiments under UV irradiation confirm the electron scavenging ability of metal NPs. The Ag/TiO₂ spectrum displays a maximum absorption peak at 460 nm (Fig. 4a). This spectrum is assigned to the combination of signals arising from the accumulation of photoholes in TiO₂[46,47] and a blue shift of the Ag plasmon absorption band due to confinement of photogenerated electrons by the generated interfacial depletion layer[48]. A bleaching signal appears in the 500–650 nm region (Supplementary Figure 9) that corresponds to the disappearance of the plasmon absorption band. Similar results have been reported in the literature for Ag@TiO₂ core–shell composites[44]. It is worth noting that the steady-state absorption of Ag NPs shows its maximum at ca. 500 nm (Supplementary Figure 8), while the transient absorption bleach of the SPR ground state appears ~150 nm red-shifted (Fig. 4b). We assign this shift to the fact that, at 500 nm, the positive-TAS signal corresponding to TiO₂ holes and the blue shift of the plasmon absorption band overlap with the negative signal of the SPR bleach, and masks the maximum absorption wavelength of that bleach. The absorption spectrum of the Ag SPR did not change during these spectroscopic measurements, confirming the stability of the sample under irradiation (Supplementary Figure 10).

To determine the effect of charge separation on the photocatalytic improvement, the lifetime of charge carriers in Ag/TiO₂, deduced from TAS measurements, were compared to that of bare TiO₂. The lifetimes of electrons and holes in titania (measured at 900 and 460 nm, Supplementary Figure 11) show identical kinetics assigned to bimolecular recombination[47,49,50] on the microsecond–millisecond timescale ($t_{50\%} \approx 50 \mu s$). The deposition of Ag NPs onto TiO₂ results in an eight fold-increased transient absorption signal amplitude of photoholes, compared to TiO₂, indicative of the accumulation of long-lived holes in the VB of the semiconductor upon electron transfer to Ag NPs (Fig. 4a). The lifetime of these holes is also increased by three orders of magnitude ($t_{50\%} \approx 0.5 s$), demonstrating the suitability of using Ag NPs to suppress the electron/hole recombination in the semiconductor. We note, however, that both direct electron transfer and recombination reactions start occurring at a faster timescale than the resolution limit of our instrument. The decay kinetics of Ag/TiO₂ photoholes exhibits primarily a monoexponential decay with a small increase in the signal amplitude at 460 nm in the $10^{-5}-10^{-2}$ s timescale (Fig. 4a). The decays of Ag/TiO₂ probed at 650 nm show an increasing negative amplitude of the bleached signal over the same range of timescales (Fig. 4a). We assign both behaviours to the kinetics of electron transfer between TiO₂ and Ag NPs ($t_{50\%} \approx 50 \mu s$), with the negative signal at 650 nm corresponding to the disappearance of the ground state plasmon absorption band, and the rise at 460 nm to the appearance of a blue-shifted absorption of the Ag plasmon band. The decrease in the photogenerated electron/hole recombination rate is also corroborated by photoluminescence (PL) studies, where the Ag/TiO₂ catalyst shows a quenching of PL that increases with silver loading (Supplementary Figure 12).

Photocurrent densities obtained in a three-electrode configuration photoelectrochemical cell at different bias potentials and under UV illumination exhibit an improvement of ca. 0.1 mA cm$^{-2}$ when Ag NPs are supported on TiO₂ (Fig. 4c). This confirms that the presence of silver NPs decreases the electron recombination and enhances the charge transfer and separation. Moreover, EIS has been performed under dark and illumination conditions at different bias potentials, and the Nyquist plots at 0 V vs. Ag/AgCl (Fig. 4d) show that the resistance of the TiO₂ space charge region decreases from 81 to $36 \pm 0.5 k\Omega$ when Ag NPs are supported on bare TiO₂. These values have been obtained through a simulation using the equivalent electric circuit depicted in the inset of Fig. 4d. The fitting results obtained at different bias potentials and under dark and illumination conditions are presented in Supplementary Figure 5 and Supplementary Table 4. These results are in good agreement with previous TAS observations, since they confirm a significant improvement in Ag/TiO₂ charge transfer compared to bare TiO₂.

Consequently, our experimental results indicate that, under UV illumination, Ag NPs in direct contact with the semiconductor can efficiently scavenge the photogenerated electrons from the TiO₂ conduction band, while the holes accumulate at the VB of the oxide. This spatial separation of charge carriers results in an increased lifetime up to the second timescale. We can observe two main effects derived from this charge separation: on the one hand, the accumulation of electrons in the Ag NPs can explain the higher formation of more electron-demanding products in Ag/TiO₂ compared to bare TiO₂. On the other hand, photogenerated holes show a lifetime of $t_{50\%}$ ~0.5 s (Fig. 4a). It has been reported that the water oxidation reaction at the surface of TiO₂ takes place in the 100 ms timescale[47]. Therefore, the charge separation induced by the electron transfer from the TiO₂ to the metal NPs is sufficient to allow water to be the electron source for the CO₂ reduction.

In summary, NAP-XPS and TAS measurements reveal the formation of C–H intermediates on Ag/TiO₂ under UV irradiation of a CO₂ and H₂O atmosphere, in good agreement with the CH₄ formation. Considering that the photoreduction of CO₂ consists in a multi-electron transfer process, limited by charge recombination, the improved activity of Ag/TiO₂ towards hydrocarbons can be explained by a decrease in the $e^-/h^+$ recombination rates, as addressed in TAS and PL studies.

One of the most interesting properties of Ag/TiO₂ interfaces is their ability to catalyse photoredox reactions under visible light. However, the pathways involved in this reaction are still under debate in the literature. A critical factor is the photoinduced charge dynamics and their effect on the reactivity and product distribution in CO₂ photoreduction. This influence can be clearly observed in the photocatalytic reaction performed using a Ag/TiO₂ catalyst under visible irradiation ($\lambda \geq 450$ nm), which leads to the production of CH₃OH, CH₄ and CO, whilst bare TiO₂ does not show any photocatalytic activity (Fig. 5a). This results contrast with the experiments performed at $\lambda \geq 400$ nm, in which both catalysts showed photoactivity towards CH₃OH, H₂ and CO, but only Ag/TiO₂ was active for the formation of CH₄. These data show that the interfacial synergy between plasmonic Ag NPs and the semiconductor leads to unexpected photocatalytic activity under visible light considering the band gap of TiO₂.

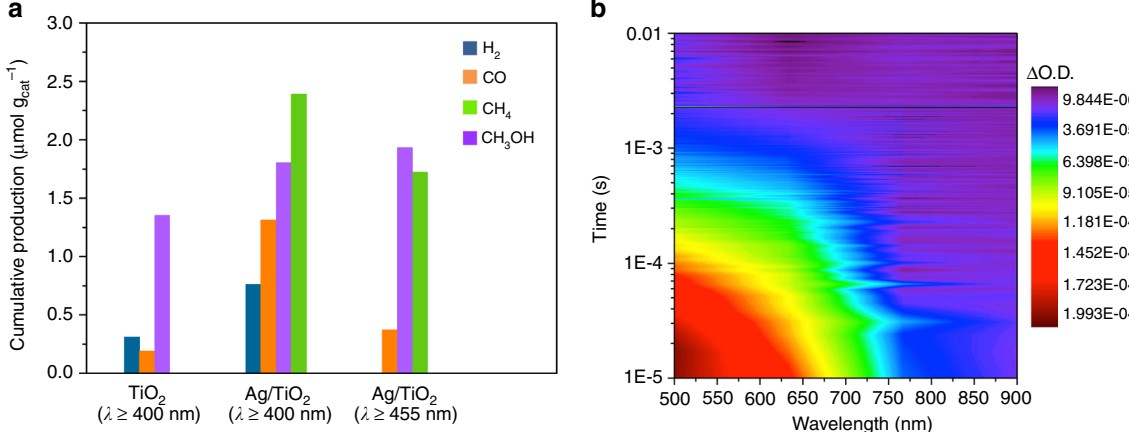

**Fig. 5** Visible-light irradiation. **a** Cumulative production (µmolg$_{cat}^{-1}$) of TiO₂ and 1.5Ag/TiO₂ in the photocatalytic reduction of CO₂ with water after 15 h under visible irradiation (cut-off filters at 400 and 455 nm). **b** Two-dimensional plot of the transient absorption spectrum of an Ag/TiO₂ film excited with visible light (510 nm, 350µJcm$^{-2}$). All experiments were performed under N₂ atmosphere

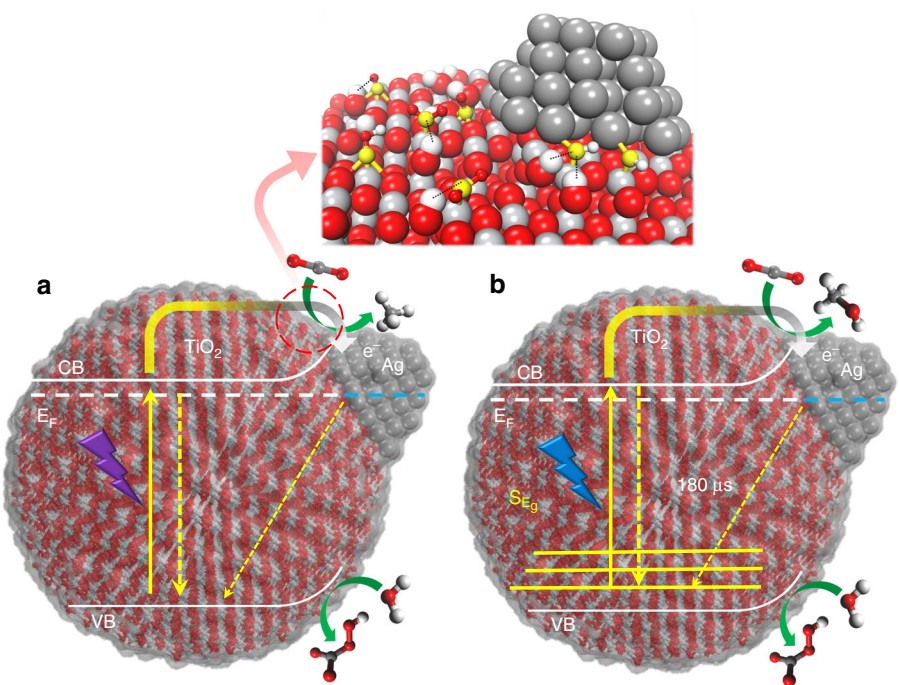

**Fig. 6** Schematic diagram of the charge dynamic processes in Ag/TiO$_2$, when exciting with **a** UV and **b** visible light. The main recombination pathways are represented with dashed arrows. CB conduction band, VB valence band, S$_{Eg}$ ub-band-gap states. Magnification of the reaction intermediates observed in this work (top). Atoms colour: titanium (light grey), oxygen (red), carbon (yellow), hydrogen (white) and silver (dark grey). Dashed lines indicate possible interaction with surface hydrogen

To explain this results, in situ UV–vis DRS spectroscopy was also used to investigate the charge transfer under visible illumination ($\lambda = 410$ nm) at the Ag/TiO$_2$ interface by following the evolution of the surface plasmon band (Supplementary Figure 13). The formation of the SPR band is clearly observed under these conditions, although with a remarkable lower intensity of two orders of magnitude compared with the experiments under UV illumination. This led to a lower photoreduction rate of Ag NPs, being necessary 8 h to achieve the complete reduction of silver. This behaviour indicates that irradiation at wavelengths closer to the TiO$_2$ band gap leads to electron transfer from surface sub-band gap states of the semiconductor to Ag NPs, although at a considerably slower rate than under UV irradiation due to the high recombination rate of photogenerated TiO$_2$ electrons[23,51].

TAS measurements under visible-light excitation ($\lambda_{ex} = 510$ nm) of Ag/TiO$_2$ reveal a small transient absorption band extended up to 750 nm (Fig. 5b and Supplementary Figure 14A), with its maximum absorption at 500 nm. This signal is enhanced when exciting at wavelengths corresponding to the maximum absorption of the Ag NPs plasmon (Supplementary Figure 15). The transient absorption decays have a $t_{50\%}$ ~180 µs (Supplementary Figure 14B), with the signal being practically zero at 10 ms, indicating that Ag NPs are not being irreversibly oxidised to Ag$^+$ upon excitation with visible light. Thus, visible-light excitation of Ag/TiO$_2$ results in shorter-lived transient absorption decay that had smaller signal amplitude compared to the one obtained when exciting the sample with UV light. This decay is assigned to a smaller generation of charge carriers, which explains the lower efficiency of visible-light-driven CO$_2$ reduction compared to UV excitation. To verify that the TAS signal arises from the interaction between the TiO$_2$ and Ag NPs, control measurements were performed. No signal in the microsecond–second timescale was observed while irradiating an aqueous Ag NPs suspension (0.02 mg mL$^{-1}$) in the absence of

TiO$_2$ under UV or visible light, or when exciting a bare TiO$_2$ film with visible light (Supplementary Figure 16).

To sum up, we have reported operando spectroscopic evidence and theoretical insights for the unique CO$_2$ photoreduction performance of Ag/TiO$_2$ under UV and visible irradiation. We show that the strong interfacial dielectric coupling between plasmonic metal NPs and TiO$_2$ leads to the formation of surface sub-band gap states, which are the key actors to promote an efficient separation of photogenerated charge carriers and a consequent enhanced selectivity towards highly electron-demanding products. Under UV irradiation, Ag/TiO$_2$ is highly selective towards CH$_4$ (Fig. 6a). This improved reactivity to methane is attributed to the electron scavenging ability of Ag NPs, which results in a lifetime of charge carriers increased by three orders of magnitude compared to bare TiO$_2$. In situ studies support not only this charge transfer but also the formation of C–H intermediates in the pathway to methane following the carbene mechanism (Supplementary Figure 17). Ag NPs are also indispensable for the visible light-induced photoactivity, predominantly towards CH$_3$OH. Our results suggest that the strong interfacial Ag–TiO$_2$ interaction induces photogenerated electronic transitions from intra-band gap surface states, allowing the formation of electron/hole pairs in the near-surface region. Shorter-lived transient absorption decay under visible excitation indicates a smaller generation of charge carriers, which explain the decrease in photoactivity and the shift of selectivity to lower electron-demanding products such as CH$_3$OH (Fig. 6b). Our studies go a step forward in the understanding of light-induced charge transfer processes on plasmonic metal/semiconductor interfaces for CO$_2$ photoreduction. The successful combination of in situ surface and time-resolved spectroscopic techniques with DFT calculations has shed light on the phenomena occurring in the photocatalytic reaction at different timescales, revealing the key factors that determine the photoactivity of Ag NPs under UV and visible irradiation. Application of the interconnected

strategies described here will enable the development of a wide range of emergent light-controlled materials and reactions.

## Methods

**Materials and synthesis.** All chemicals were purchased from commercial suppliers and used without further purification. Anatase-type $TiO_2$ was obtained from Crystal ACTIV™ (PC500, Lot Number 6293000586), and it was stabilised with a thermal treatment at 400 °C for 4 h prior to use. A commercial sample was selected in order to minimise the experimental error and perform systematic measurements with good reproducibility.

Ag/$TiO_2$ powdered catalysts were prepared by a wet impregnation procedure. An aqueous $AgNO_3$ solution (0.5–3.0 Ag wt%) was added to a conical flask containing commercial anatase-type $TiO_2$. The mixture was mixed in a rotary evaporator at 40 °C until water was completely removed. The resulting product was dried at 100 °C for 13 h, ground and calcined in air at 400 °C for 4 h. In order to ensure the complete reduction of Ag, freshly prepared Ag/$TiO_2$ NPs were irradiated for 15 min with UV light (365 nm, 10 W deuterium lamp).

For spectroscopic measurements, analogous nanocrystalline Ag/$TiO_2$ films were employed. Firstly, anatase $TiO_2$ films were prepared from a colloidal paste by the Doctor Blade technique as reported previously[52]. Films were dried for 10 min before being calcined in air at 450 °C during 30 min, and cut in small pieces of approximately 1 cm × 1.5 cm. The resulting thicknesses were 4 μm, measured by profilometry (Tencor Instruments). Ag/$TiO_2$ films were prepared by coating Ag NPs over the $TiO_2$ films by photodeposition of an aqueous $AgNO_3$ solution (0.1 M). The silver content was adjusted by varying the irradiation time with a xenon lamp (75 W) in a range of 5–60 s. To ensure that all $Ag^+$ was reduced to Ag, the films were irradiated for 1 h under a $N_2$ atmosphere with a Nd-YAG laser (Big Sky Laser Technologies Ultra CFR) with a wavelength of 355 nm (1 mJ cm$^{-2}$, 10 Hz, 6 ns pulse width). A commercial aqueous dispersion of Ag NPs (10 nm, 0.02 mg mL$^{-1}$) was used as a control in spectroscopic measurements.

**Photocatalytic $CO_2$ reduction tests.** Experiments were conducted in continuous-flow mode in a home-made reaction system. The powdered catalyst (0.1 g) was deposited on a glass microfiber. Illumination was carried out using four 6 W lamps with a maximum wavelength at 365 nm and an average intensity of 47.23 W m$^{-2}$ (measured by a Blue-Wave spectrometer in the range 330–400 nm). A 30 W white light-emitting diode (LED), with cut-off filters at 400 or 455 nm, was used for visible-light illumination (average intensity 53.89 W m$^{-2}$ measured in the region 400–600 nm). Compressed $CO_2$ (≥99.9999%, Praxair) and water (Milli-Q), were passed through a controlled evaporation mixing unit, maintaining a molar ratio of 7.25 ($CO_2$:$H_2O$). The reaction conditions were set at 2 bar and 50 °C. In-line gas chromatography (Bruker 450-GC) analyses were performed to detect the reaction products. The GC is equipped with two separation branches, one equipped with two semicapillary columns (BR-Q Plot and BR-Molesieve 5A) and one thermal conductivity detector, a flame ionisation detector (FID) and a methanizer. The second separation branch consists on a capillary column (CP-Sil 5B) and a second FID. Before starting the experiments the reactor was first degassed under vacuum and then purged for 1 h using argon (100 mL min$^{-1}$) to remove any residual organic compounds weakly adsorbed to the surface of the catalyst. Then, the reactor was flushed with the $CO_2$ and water mixture for 1 h to establish an adsorption–desorption balance at the reaction temperature.

**General characterisation.** The silver content of the catalysts was determined by inductively coupled plasma–optical emission spectrometry analyses with a Perkin Elmer Optima 3300 DV instrument by digesting the solid in a mixture of HF and $HNO_3$. This content was slightly lower than the nominal concentration due to a loss of metal during the washing step of the synthesis. Specific surface areas were calculated from $N_2$ adsorption–desorption isotherms at 77 K measured on a QUADRASORB instrument. The samples were degassed under vacuum at 105 °C for 20 h in $N_2$ before the measurement. Crystal structures were characterised by X-ray diffraction that was performed using a Philips PW 3040/00 X'Pert Multi-Purpose/Materials Research diffractometers with Cu Kα radiation (λ = 1.54178 Å) at a scanning rate of 0.2 ps$^{-1}$. Morphology studies of the particles was performed using a Philips Technai 20 Transmission Electron Microscope, operating with a tungsten filament working at 200 kV. To perform the measurements, the powdered samples were suspended in acetone and a drop of that suspension was deposited on a copper microgrid. Raman spectra of the catalysts, before and after reaction, were recorded at room temperature using a JASCO NRS-5000/7000 series spectrometer with an excitation wavelength of 532 nm. PL experiments were carried out with a Fluorescence Spectrometer Perkin Elmer LS 55, using an excitation wavelength of 300 nm and a cut-off filter at 350 nm.

**Theoretical calculations.** Theoretical calculations by periodic DFT were carried out using an $Ag_9$ cluster with fcc structure and surface formed by (111) planes. Initially Ag atoms were connected to O bridge atoms of a 114 atom $TiO_2$ NP with anatase structure[53]. Geometry and electronic structure were performed using the projected augmented wave method implemented in Viena ab initio simulation package[54,55]. The total energies corresponding to the optimised geometries of all samples were calculated using the spin polarised version of the

Perdew–Burke–Ernzerhof[56]. In addition, the Heyd–Scuseria–Enzerhof hybrid functional (HSE06)[57], with a mixing parameter of 0.325, was used to fit a more accurate energy gap. The cut-off for the kinetic energy of the plane-waves was set to 450 eV to ensure a total energy convergence better than $10^{-4}$eV. The cut-off for the kinetic energy of the plane-waves was set to 500 eV to ensure a total energy and force convergence better than $10^{-4}$eV and 0.01 eV/Å$^3$, respectively. The VESTA package v.3.3.9 was used to represent the electron localisation[58].

**Electrochemical and photoelectrochemical measurements.** Electrochemical and photoelectrochemical measurements were performed in a three-electrode glass cell with a quartz window containing an aqueous solution of 0.5 M $Na_2SO_3$ with a pH=9. $TiO_2$ and Ag/$TiO_2$ powders suspensions were deposited by drop casting on indium tin oxide glass and used as working electrode. The counter electrode was a platinum wire, and the reference one was a Ag/AgCl electrode. Voltage, current density (at dark and under illumination) and EIS were measured with a potentiostat–galvanostat PGSTAT204 provided with an integrated impedance module FRAII (10 mV of modulation amplitude is used at 400 Hz). A UV LED lamp was used as light source. Measured light power density reaching the surface sample is 80.4 ± 0.2 mW cm$^2$.

**Near-ambient pressure X-ray photoelectron spectroscopy.** In situ NAP-XPS was measured with a PHOIBOS 150 NAP energy analyser at the CIRCE beamline of the ALBA synchrotron light source. The CIRCE beamline is an undulator beamline with a photon energy range 100–2000eV. In this work, data were acquired with photon energy $hv$ = 900 eV, using an analyser pass energy of 20 eV. The beam spot size at the sample is 100 × 20 μm$^2$. Powdered samples were pressed into round disks. Ag 3d NAP-XPS spectra were recorded in UHV and under both dark and illumination conditions. In addition, C 1s and O 1s were acquired in UHV and under a $CO_2$ and $H_2O$ mixture. NAP-XPS experiments under illumination were performed using a high power UV LED (Hamamatsu Co.) with a maximum emission centred at 365 nm. A Au foil was taken as reference. These studies were performed as follows: ultrapure water (LC–MS CHROMASOLV® grade, Sigma-Aldrich) was introduced into the chamber through a variable high precision leak valve after being degassed by multiple freeze–pump–thaw cycles. $CO_2$ (purity ≥ 99.995%, Abelló Linde) was introduced into the chamber directly from a commercial cylinder. The gases were admitted in the chamber using a leak valve until reaching a base pressure of $5 \times 10^{-2}$mbar ($4.375 \times 10^{-2}$mbar and $6.250 \times 10^{-3}$mbar of $CO_2$ and $H_2O$, respectively). Data were collected using the SpecsLab software.

**Light-induced charge transfer measurements.** UV–vis DRS of powdered samples were obtained by a Perkin Elmer Lambda 1050 UV/vis/near-infrared spectrometer using a deuterium lamp (10 W, λ = 200-400 nm) and a tungsten halogen lamp (30 W, λ > 400 nm). Reduction kinetic studies were performed by irradiating Ag/$TiO_2$ samples with UV (LED 1 W, λ = 369 nm) and visible light (10 W lamp, λ = 410 and 500 nm) for different periods of time.

The microsecond–second transient absorption decays were measured using a home-built system as reported elsewhere[26]. The third harmonic of a Nd:YAG laser (Big Sky Laser Technologies Ultra CFR Nd:YAG laser system, 6 ns pulse width, $\lambda_{ex}$ = 355 nm) was used to excite Ag/$TiO_2$ samples with UV light. A commercially available optical parametric oscillator (Oppolette) pumped by an Nd:YAG laser was used to excite the sample in the range of 410–700 nm (pulse width < 20 ns). The laser intensity was adjusted using neutral density filters as appropriate, with experiments typically employing 350 μJ cm$^{-2}$. The decays observed were the average between 500 and 1000 averages laser pulses. The data were processed using home-made software based on Labview. Prior to each TAS measurement, samples were degassed with $N_2$ and irradiated for 1 h with 355 nm UV light (1 mJ cm$^{-2}$).

Bare $TiO_2$ and Ag/$TiO_2$ PL spectra show a broad band from 350 to 600 nm (3.5–2.1 eV) composed by multiple emission contributions (Supplementary Figure 12). The UV emission peak at ca. 396 nm is attributed to the band-to-band radiative recombination process with wavelength close to the band gap of anatase ($E_g$ = 3.2 eV; 387 nm).

## Data availability
The data sets within the article and Supplementary Information of the current study are available from the authors upon request.

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

## Acknowledgements

This work has received funding from the European Research Council (ERC), within the projects Intersolar (291482) and HyMAP programme (648319) under the European Union's Horizon 2020 research and innovation programme. The results reflect only the authors' view and the Agency is not responsible for any use that may be made of the information they contain. V.A.P.O. thanks funding from the Spanish Ministry of Economy and Competitiveness (project ENE2016-79608-C2-1-R). A.R. acknowledges the European Commission Marie Curie CIG. M.B. acknowledges the Juan de la Cierva Formación program: FJCI-2016-30567, V.A.P.O. acknowledges support from the Centre of Supercomputacio de Catalunya (CESCA) and to ALBA Cells Synchrotron facilities. Authors thank Dr. Xiaoe Li for assistance in TiO$_2$ films preparation. Support from the Repsol Technology Centre in gratefully acknowledged.

## Author contributions

L.C. synthesised and characterised the samples, and carried out the CO$_2$ photocatalytic experiments. A.R. and J.D. designed and carried out the TAS measurements. L.C., F.F., J. M.C. and V.A.P.O. performed the in-situ characterisation and made the NAP-XPS measurements, in collaboration with V.P.D. and C.E. M.B. performed the electro- and photoelectrochemical characterisation. V.A.P.O. made the theoretical calculations and mechanistic studies. V.A.P.O. and D.P.S. conceived and designed the experiments. All authors discussed the results and commented the manuscript.

## Additional information

**Competing interests:** The authors declare no competing interests.

