## [Peer Review File · Nature Communications]

Reviewers' comments:

Reviewer #1 (Remarks to the Author):

A really nice and thorough work on the Ag/tiO₂ system used in CO₂ photoreduction as well as a work towards molecular and atomic level understanding the mechanisms and processes behind. I am highly recommend this paper but have a couple of main concerns and smaller ones either:

1. Authors are combining molecular level understanding techniques from the intermedier side (NAP-XPS) as well as from the electronic strucutre side (PCT, TAS etc.). However, im not exactly see the relationship between them. Its interesting that under UV CH₄ is the main product and under visible light a lot of methanol is forming, where the explanation is really deep and nice, but the intermedier explored by NAP-XPS is not exactly adding new informations. Which is okay, but one of the main conclusions of the manuscript at line 264-274 mainly based on the synergy of the used techniques. Line 274 speaks about drawing a global mechanism. Please add the mechanism or reaction routes to this paper. It would be great if authors can combine the photoreduction and plasmonic knowledge with the CO₂ reduction mechanistis studies (from heterogeneous catalysis eg.: Tuning Selectivity of CO₂ Hydrogenation

Reactions at the Metal/Oxide Interface Shyam Kattel, Ping Liu, and Jingguang G Chen J. Am. Chem. Soc., DOI:

10.1021/jacs.7b05362) and come up with a reaction route.

2. Before arranging the mechanisms, In NAP-XPS evaluation (Fig. 3.), please check on other articles (catal. lett. 2018, 148:1686 and B. Lesiak et al. / Journal of Electron Spectroscopy and Related Phenomena 193 (2014) 92–99), and reconsider your claims over there and please show in a textbox on the graphs the used pressures and gas concetrations. Maybe you can get deeper informations about the intermediers on the surface.

3. As insert of Fig. 3. shows, the silver is reducing continously, your system is dynamic. Its always changing what you are tried to follow with a couple of techniques. How the system, CO₂ reduction mechanism can change with time?

Smaller ones:

1. Where is the Ag NP on Figure S2/D,G? How could you make size distrubuiton for them?

2. I guess 3 % of Silver could be detectable by XRD at this sizes, of course its not obligatory, just please check.

3. Line 69 - 73: why the 1.5 % sample is the best? Why are these catalysts are soo good? What is your technique? Also, it will be good to somehow show that under Uv light there are more CH₄ but VIS produces methanol.

4. Line 129 in SI: For spectroscopy the prepared sample was a bit differetn from ones used for catalysis. Are these samples are comparable totally? Any differences?

5. line 94: In the future, just a suggestion to try to use EXAFS to investigate Ag-O bonds.

6. line SI 103: Please check the pressures and the sum of them. Also, im not exactly sure how big was the pressure. please show it on fig. 3.

7. Line 150: Authors says that we should see O₂. Why?

8. SI Table 2. and figure 5.A is somehow connecting to each others. I mean that is also a great result (with molecular level understanding) next to the 15 fold increment that UV light producing more methanol compared to UV lighted samples.

Reviewer #2 (Remarks to the Author):

The manuscript by Collado et al. reports a comprehensive investigation of charge transfer in Ag/TiO₂ photocatalysts for CO₂ reduction in an aqueous solution. As a result of numerous experimental measurements and theoretical investigations, the authors report that the electron scavenging ability of Ag nanoparticles, facilitated by the formation of surface sub-band gap states in the TiO₂, help push the selectivity of the reaction toward highly electron-demanding products under UV illumination. This selectivity is somewhat lost upon visible illumination as a result of the faster decay of transient absorption suggesting the fewer carriers are photogenerated.

I am grateful that the authors summarized these findings in the end, because it was difficult to reach these conclusions during the course of the manuscript. The authors have previously published the result that Ag nanoparticles scavenge electrons from TiO₂ (refs. 17,25), so this key finding has already been reported. That this may be exploited to select for electron-starved products is a natural conclusion one may draw. What, then is the new result that is being reported here? Their confirmation of this conclusion through extensive measurements and modeling?

If so, this manuscript is not appropriate for NatComm, not only because the findings are too incremental, but also because the manuscript length is too short for the authors to provide sufficient detail and explanations for each of these confirming measurements and analyses. I simply do not understand Figures 2, 4, and 5B; the affiliated discussion repeatedly refers only to each sub-figure and draws a conclusion that is not sufficiently explained or motivated. Perhaps each conclusion may be inferred by the few readers who are experts in a given technique, but the authors use so many different types of techniques that it is very unlikely someone could read this article with sufficient breadth of knowledge in all these areas to infer all these conclusions. (I'm particularly confused about the basis for which the sub-band gap states are deduced, ostensibly a key finding of this manuscript. I'm also not sure I understand how faster TA decay means fewer carriers generated.) The authors seem to recognize this, as the SI for this manuscript is enormous, 23 pages in all, and frequently cited in the manuscript.

If the authors objective was to extract the key findings and summarize them in a short, letter-style format, I'm afraid the attempt failed. Indeed, the attempt may be impossible, given the breadth and depth of the actual work done and the discussions that are required to draw it all together. A much more compelling article would have merged the manuscript and SI into a more thorough explanation of how all these pieces fit together to tell a coherent, compelling story, perhaps for a high impact journal like ACS Nano.

Reviewer #3 (Remarks to the Author):

This manuscript reports a wide and complex characterization of Ag-TiO₂ system for the CO₂ photoreduction reaction. This reaction is actually a hot topic in photocatalysis. Up to now, the reported results demonstrate that this a challenging reaction. So, hard investigations about the catalyst

mechanism is really necessary. For this reason, the opportunity of this manuscript is clear and deserve to be considered in this journal.

However, due to the high number of techniques used, a more concise and clear discussion is needed for communication paper. Some arguments appears repeated along the text.

As a general comment, I would recommend authors to lighten the discussion in some sections, especially "Photoinduced charge transfer" section is somehow confusing. Authors should check the numbering of Figures in this section.

The conclusion of the MS is at the end is not clear. The key point is the different photocatalytic behavior of Ag-TiO₂ under UV and vis conditions leading to different products in each case. Authors should emphasize this point as conclusion.

At the end of the MS, the "Methods" section is quite redundant since some parts are repeated in the Supplementary Information. This part should be placed at SI.

In Table 3 at SI, CO evolution rate at the first row is not correct, should be 0.7 micromol/g.h. The actual value corresponds to the CO production (11.1 micromol after 15 h) as it is shown in Table 2. After these considerations, I would recommend the publication of this MS in Nature Communication.

Reviewers' comments:

Reviewer #1 (Remarks to the Author):

Comments:

A really nice and thorough work on the Ag/TiO₂ system used in CO₂ photoreduction as well as a work towards molecular and atomic level understanding the mechanisms and processes behind. I am highly recommend this paper but have a couple of main concerns and smaller ones either:

1. Authors are combining molecular level understanding techniques from the intermedier side (NAP-XPS) as well as from the electronic structure side (PCT, TAS etc.). However, im not exactly see the relationship between them. Its interesting that under UV CH₄ is the main product and under visible light a lot of methanol is forming, where the explanation is really deep and nice, but the intermedier explored by NAP-XPS is not exactly adding new informations. Which is okay, but one of the main conclusions of the manuscript at line 264-274 mainly based on the synergy of the used techniques. Line 274 speaks about drawing a global mechanism. Please add the mechanism or reaction routes to this paper. It would be great if authors can combine the photorediction and plasmonic knowledge with the CO₂ reduction mechanistis studies (from heterogeneous catalysis eg.: Tuning Selectivity of CO₂ Hydrogenation Reactions at the Metal/Oxide Interface Shyam Kattel, Ping Liu, and Jingguang G Chen J. Am. Chem. Soc., DOI: 10.1021/jacs.7b05362) and come up with a reaction route.

Response: We appreciate the reviewer comments and the literature suggestion. We try to clarify below the synergy between the spectroscopic techniques used here and their role to ascertain the reaction pathway and the involved reaction intermediates.

In-situ NAP-XPS are essential to ascertain the first step of the CO₂ adsorption/activation, via carbonate/bicarbonate formation. These measurements also reveal the formation of methylene and carbonyl intermediates prior to the formation of hydrocarbons. In addition, Raman spectroscopy shows the formation of peroxo and peroxocarbonate species, with an important implication on the oxygen evolution. However, these measurements do not explain the distinct behaviour of Ag/TiO₂ on the preferential formation of hydrocarbons (especially methane), in comparison to TiO₂ (major formation of CO). Thus, TAS measurements demonstrate that Ag NPs promote an efficient separation of photogenerated charge carriers, significantly reducing the electron/hole recombination rates and shifting the reaction selectivity towards highly electron demanding products.

Regarding the reaction mechanism, the results obtained herein suggest that the CO₂ photoreduction follows a formaldehyde reaction pathway under our experimental conditions. This has been included in the "Mechanistic studies of CO₂ photoreduction" section of the revised manuscript along with proper references:

“The most relevant finding in the C 1s spectrum is the appearance of two new components at 284.0 eV and 287.0eV, corresponding to a methylene and formate intermediates, respectively (Figure 3A).^{33,34}”

“During the reaction a decrease in OH groups and physisorbed H₂O in O 1s spectrum was observed (Figure 3B) indicating the role that both are playing in the photoredox process. In summary, NAP-XPS observations in Ag/TiO₂ reveal an initial CO₂ adsorption/activation step via carbonate/bicarbonate species followed by the formation of methylene and formate species, which are identified as the intermediates of hydrocarbons tentatively through the formaldehyde reaction pathway.³⁸”

“Raman studies (Supplementary Figure 7) show that the fresh Ag/TiO₂ surface is covered by different carbon species adsorbed from ambient air. The observed bands are assigned to linear and bent adsorbed CO₂, bicarbonate as well as bidentate and monodentate carbonate.³⁹ After reaction, some changes are observed, mainly related to a significant increase of adsorbed CO₃²⁻ and HCO₃⁻ species and the appearance of formate (1366 cm⁻¹) as possible reaction intermediate⁴⁰ in agreement with with NAP-XPS studies. The increase in the bands due to bent CO₂ structures may be the consequence of the generation of CO₂⁻ species coordinated to metal ions. It is worth highlighting too the appearance of a new Raman band at a shift of ca. 910 and 985 cm⁻¹ assigned to the ν_{O-O} vibrational mode of surface peroxo Ti(O₂)⁴¹ and peroxocarbonate species (CO₄²⁻ and C₂O₆²⁻)^{39,42} respectively. This observation suggests that the formation of both peroxidized species arise, on the one hand, from hole capture by CO₃²⁻ and HCO₃⁻ and, on the other hand, from partial oxidation of water. This could prevent O₂ from being detected as the outcome of H₂O oxidation, rendering the oxygen balance a troublesome task in this reaction. “

In addition, we have modified the figure 6 including the reaction intermediates.

Figure 6. Schematic diagram of the charge dynamic processes in Ag/TiO₂, when exciting with (A) UV light and (B) visible light. The main recombination pathways are represented with dashed arrows. CB (Conduction Band); CV (Valence Band); S_{Eq}= Sub Band-Gap states. Magnification of the reaction intermediates observed in this work (top). Atoms Colour: Ti (Light grey), Oxygen (red), Carbon (yellow), Hydrogen (white), Silver (dark grey). Dashed lines indicate possible interaction with surface hydrogen.

2. Before arranging the mechanisms, In NAP-XPS evaluation (Fig. 3.), please check on other articles (catal. lett. 2018, 148:1686 and B. Lesiak et al. / Journal of Electron Spectroscopy and Related Phenomena 193 (2014) 92–99), and reconsider your claims over there and please show in a textbox on the graphs the used pressures and gas concentrations. Maybe you can get deeper informations about the intermediers on the surface.

Response: Thanks again for your comments. Figure 3 has been amended in the revised manuscript as suggested by the reviewer, including the gas pressures and the CO₂:H₂O molar ratio in the figure caption.

Figure 3. In-situ NAP-XPS spectra of C1s (A) and O1s (B) signals recorded for Ag/TiO₂ under the following conditions (from bottom to top): UHV; after dosing CO₂ and H₂O ($P_{CO_2} = 4.375 \cdot 10^{-2}$ mbar and $P_{H_2O} = 6.250 \cdot 10^{-3}$ mbar ; $P_{Total} = 5 \cdot 10^{-2}$ mbar); and exposed to UV irradiation (365 nm). The inset displays the NAP-XPS experimental set-up, illustrating the photochromism of Ag NPs (Ag⁺ to Ag⁰ reduction under UV illumination): (1) Catalyst; (2) sample holder; (3) analyzer.”

In addition, the literature suggested by the reviewer has been taken into consideration. In particular, further reaction intermediates have been identified, such as formate species at 287.0 eV and we have included the Catal. Lett. reference [34]

“The most relevant finding in the C 1s spectrum is the appearance of two new components at 284.0 eV and 287.0eV, corresponding to a methylene and formate intermediates, respectively (Figure 3A).^{33,34}”

3. As insert of Fig. 3. shows, the silver is reducing continuously, your system is dynamic. Its always changing what you are tried to follow with a couple of techniques. How the system, CO₂ reduction mechanism can change with time?

We agree with the reviewer comment, Ag NPs reduction is a dynamic processes that occurred progressively over a period of time depending on the irradiation source properties (intensity, wavelength and irradiation time). This progressive reduction is evident from the gradual darkening of the sample (inset of Figure 3). However, this system is not permanently dynamic and it reaches a steady state that depends on the aforementioned conditions. In our case, Supplementary Figure 8 shows the increase in the intensity of the SPR absorption band under UV irradiation (365 nm), which reaches a steady state after 5 min of irradiation. As time passes, the plasmonic band does not show any significant change, remaining the same after

15 min (Supplementary Figure 8). This indicates a steady state of the photoreduction of Ag NPs is reached. This doesn't mean that all silver is reduced only that a photoreduced/oxidised equilibrium is achieved. On the other hand, when samples are irradiated under visible light ($\lambda = 410$ nm) Ag NPs are progressively reduced at a considerably lower rate than under UV irradiation, reaching the steady state after 8 h of visible illumination (Figure R1).

A second indirect evidence that system reach this steady state is with the transient absorption measurements. Each measurement takes between 8 and 16 min. Repeated measurements of the same sample showed identical electron transfer kinetics between the TiO_2 and the Ag NPs.

Figure R1. Temporal evolution of the UV/Vis spectra of 1.5Ag/TiO₂ sample under visible illumination ($\lambda = 410$ nm).

Smaller ones:

1. Where is the Ag NP on Figure S2/D,G? How could you make size distribution for them?

Response: The reviewer is right pointing out that Ag NPs are difficult to distinguish in images D and G in Figure S2, due to the low Ag content and particle size. Below we include a sample of some TEM images that we used for the particle counting, in which Ag NPs are marked in red (Figure R2).

Figure R2. TEM images of 0.5Ag/TiO₂ (I,II) and 1.0Ag/TiO₂ (III,IV) samples and their corresponding Ag NPs (red squares).

2. I guess 3 % of Silver could be detectable by XRD at this sizes, of course its not obligatory, just please check.

Response: Silver could not be detected in XRD analyses, as shown in Supplementary Figure 1. We include below a magnification of the XRD spectra of 3.0Ag/TiO₂ sample (Figure R3). Here, silver oxide phases were not found, probably due to the small size (<2.5 nm) of Ag NPs. In the case of metallic silver, XRD reflection could overlap with anatase diffraction ($2\theta = 38.1^\circ$), thus not being visible in the present analysis.

Figure R3. XRD patterns of 3.0Ag/TiO₂ sample (A) and detail of region 2 θ = 32-40° (B).

3. Line 69 - 73: why the 1.5 % sample is the best? Why are these catalysts are soo good? What is your technique? Also, it will be good to somehow show that under Uv light there are more CH₄ but VIS produces methanol.

Response: Ag/TiO₂ catalysts were prepared following a deposition-precipitation method, using AgNO₃ as metal precursor (further details can be found in Supplementary Section 1.1). The catalyst 1.5Ag/TiO₂ showed the highest performance in terms of hydrocarbon production (Supplementary Table 2). In our work, we observed that the increase in Ag content led to an improved CO₂ photoreduction performance up to 1.5 wt.% loading, while hydrocarbon production decreased above that metal content. The different behaviour of 3.0Ag/TiO₂ sample was ascribed to different factors. First, 3.0Ag/TiO₂ sample shows the lowest BET area of the series (Supplementary Table 1), which may negatively affect the catalytic activity since a lower surface area implies a lower content of surface active sites. Besides, 3.0Ag/TiO₂ presents the highest silver particle size of the series (Supplementary Table 1) with a certain proportion of Ag NPs larger than 2 nm (Supplementary Figure 20). Finally, higher proportions of silver can lead to a shadow effect diminishing the UV light that reach to the TiO₂ and decreasing the photoactivity. In general, smaller NPs deposited over semiconductors are able to induce a higher shift of the Fermi level towards negative potentials than large particles, which is also related to more efficient charge transfer processes and more reductive systems. In addition, larger Ag NPs may form aggregates more easily that can act as recombination centres, thus decreasing the photocatalytic activity of the system. All these factors are believed influence the decrease in the catalytic activity observed for 3.0Ag/TiO₂ sample. We have include a paragraphs in the main text.

“The increase in the silver loading leads to a hydrocarbon production decrease that could be ascribed to different factors such as a lower Ag dispersion and surface area (Supplementary Table 1 and Supplementary Figure 20), which also provoke a shading effect that hinders the light absorption by the semiconductor.”

4. Line 129 in SI: For spectroscopy the prepared sample was a bit different from ones used for catalysis. Are these samples comparable totally? Any differences?

Response: All spectroscopic measurements were carried out using the same powder catalysts as the one used for catalysis, except for TAS analyses. In this case, according to the analysis requirements, measurements were performed using Ag/TiO₂ and TiO₂ films. These films were prepared trying to mimic as much as possible the powdered samples. They were prepared using TiO₂ anatase and Ag NPs were deposited on top using the same precursor and nominal metal loading as in powdered catalysts. Therefore, it can be concluded that both powdered catalysts and nanocrystalline films prepared for this study are totally comparable.

5. line 94: In the future, just a suggestion to try to use EXAFS to investigate Ag-O bonds.

Response: We really appreciate the reviewer's suggestion and we will submit a synchrotron proposal to evaluate the Ag-O bonds.

6. line SI 103: Please check the pressures and the sum of them. Also, im not exactly sure how big was the pressure. please show it on fig. 3.

Response: This is a mistake in the Supplementary Section 6 and we thank the reviewer for noticing it. The sum of gas pressure is $5 \cdot 10^{-2}$ mbar. As suggested by the reviewer, CO₂ and H₂O pressures have been included in the Figure 3 caption.

7. Line 150: Authors says that we should see O2. Why?

Response: In this work, H₂O is used as the electron donor to carry out the CO₂ photoreduction process. Thus, H₂O needs to be oxidised to form O₂. Therefore, O₂ should be released in the absence of any other oxidation process. However, NAP-XPS and Raman analyses revealed the presence of peroxo and peroxocarbonate species, which are formed from hole capture by CO₃²⁻ and HCO₃⁻ and from partial oxidation of water. The consumption of oxygen to form those species explains why O₂ is not detected in GC analyses.

8. SI Table 2. and figure 5.A is somehow connectiong to each others. I mean that is also a great result (with molecular level understanding) next to the 15 fold increment that UV light producing more methanol compared to UV lighted samples.

Response: We thank the reviewer for this comment. In the revised manuscript, we have tried to make clearer the unique behaviour of Ag/TiO₂ catalysts in UV and visible-light driven CO₂ photoreduction. Therefore, we have tried to emphasize the different selectivities and photoactivities, and the key factors that determine those differences under both excitation conditions.

“One of the most interesting properties of Ag/TiO₂ interfaces is their ability to catalyse photoredox reaction under visible light. However, the optoelectronic pathways involved in this reaction are still under debate in the literature. A critical factor is the photoinduced charges dynamics and their effect on the reactivity and product distribution of CO₂ photoreduction. This influence is clearly observed in the reaction performed using an Ag/TiO₂ catalyst under visible irradiation ($\lambda \geq 450$ nm), which led to the production of CH₃OH, CH₄ and CO, while bare TiO₂ did not show any photocatalytic activity (Figure 5A)” _____

Reviewer #2 (Remarks to the Author):

Comments:

The manuscript by Collado et al. reports a comprehensive investigation of charge transfer in Ag/TiO₂ photocatalysts for CO₂ reduction in an aqueous solution. As a result of numerous experimental measurements and theoretical investigations, the authors report that the electron scavenging ability of Ag nanoparticles, facilitated by the formation of surface sub-band gap states in the TiO₂, help push the selectivity of the reaction toward highly electron-demanding products under UV illumination. This selectivity is somewhat lost upon visible illumination as a result of the faster decay of transient absorption suggesting the fewer carriers are photogenerated.

Response: We thank the reviewer for their comments. We would like to point out that the aim of the present work is to rationalize the influence of photo-charge dynamics of surface plasmon nanoparticles for gas-phase CO₂ photoreduction at the metal/semiconductor interface. The selection and development of gas-phase reaction processes rather than a liquid phase system is based on the following aspects: 1) A liquid phase reactor problems with the diffusion and the limited solubility of CO₂ in water, 2) A great part of the radiation is absorbed by the water and/or by other sacrificial agents present in the liquid phase system, which decreases the CO₂ activation rate, 3) Regarding the reactor design, it is much easier to control the fluid-dynamics variables in the case of gas phase systems. In addition, open a way for the future development of these technologies using CO₂ and H₂O directly captured from the atmosphere.

I am grateful that the authors summarized these findings in the end, because it was difficult to reach these conclusions during the course of the manuscript. The authors have previously published the result that Ag nanoparticles scavenge electrons from TiO₂ (refs. 17,25), so this key finding has already been reported. That this may be exploited to select for electron-starved products is a natural conclusion one may draw. What, then is the new result that is being reported here? Their confirmation of this conclusion through extensive measurements and modeling? If so, this manuscript is not appropriate for NatComm, not only because the findings are too incremental, but also because the manuscript length is too short for the authors to provide sufficient detail and explanations for each of these confirming measurements and analyses.

Response: References 17 and 25 have been renumbered to 16 and 24 in this revised manuscript. The reviewer has pointed out that our previous works (references 16 and 24) deal with Ag NPs scavenging electrons from TiO₂. Even though reference 16 entails gold nanoparticles, we agree that reference 24 also deals with the photoreduction of CO₂ using Ag NPs. However, both the materials and the scope of the paper differ from the ones reported here. First, in our previous study, the synthesis of Ag/TiO₂ samples was carried using a different type of TiO₂ (G5 instead of PC500). This determines not only the Ag particle size, morphology

and dispersion, but also the textural and morphological properties of TiO₂ which consequently have an impact on their physicochemical, optoelectronic and reactivity properties. Secondly, our previous work reports a catalytic study on Ag/TiO₂ and Ag/ZnO catalysts under UV irradiation. We agree with the reviewer that the enhancement of hydrocarbon production is attributed to the electron scavenging of Ag NPs. However, this paper is mainly focused on the study of the catalytic performance (did not report the electron transfer kinetics at the metal/SC interface). Whereas the motivation of the present work is the deep understanding of the effect of plasmonic resonance on the photo-induced charge transfer processes in metal/semiconductor interfaces, that are still under debate [*Nat. Photonics* 11, 806–812 (2017); *Light Sci. Appl.* 5, e16017 (2016); *Science* 349, 632 LP-635 (2015); *Nat. Photonics* 8, 95–103 (2014); among others], and on their catalytic behaviour CO₂ photoreduction under UV and visible light. Taking into account the complexity of the involved processes (that occur at different timescales), it is necessary to combine the unique in-situ characterization and theoretical tools that have been selected to perform this study.

Thanks to the use of these techniques, we are able to provide a clear picture of the overall process, from the photogeneration of charges and their separation and transfer kinetics to the identification of the reaction intermediates. These aspects are correlated to understand the CO₂ photoreduction reaction under visible and UV irradiation over Ag/TiO₂ catalysts. For all these reasons, we consider that our work provides valuable and new insights on into charge photogeneration and transfer kinetics and also these finding are linked with the CO₂ photoreduction pathways over plasmonic metal/semiconductor interfaces, which can be of interest for Nature Communications' audience.

I simply do not understand Figures 2, 4, and 5B; the affiliated discussion repeatedly refers only to each sub-figure and draws a conclusion that is not sufficiently explained or motivated. Perhaps each conclusion may be inferred by the few readers who are experts in a given technique, but the authors use so many different types of techniques that it is very unlikely someone could read this article with sufficient breadth of knowledge in all these areas to infer all these conclusions.

Response: To clarify the discussion of the mentioned figures we have include a more detailed explanation highlighted in yellow:

Regarding to figure 2:

“The combination of XPS analysis at the valence band (VB), electrochemical measurements (Mott-Schottky) and theoretical calculations allow to draw a complete scheme of the electronic properties in the metal/semiconductor interface (Figure 2)”

“As conclusion of these studies, valence band studies and DFT calculations show that Ag/TiO₂ interfaces present a complex electronic band structure leading to the formation of IFS states due to the charge donation between surface Ti-O and Ag

neighbours. In addition, this interaction also leads to changes in carrier density and in the band bending.”

Regarding to figure 4:

“The comprehension of photoinduced charge dynamics is vital to understand the subsequent photoredox reactions. This understanding involves the integration of optical and electronic processes that occur at different time scales by the use of a combination of time-resolved spectroscopies and photoelectrochemical characterization techniques (Figure 4).”

“We assign this shift to the fact that, at 500 nm, the positive TAS signal corresponding to TiO_2 holes and the blue shift of the plasmon absorption band overlaps with the negative signal of the SPR bleach, and masks the maximum absorption wavelength of that bleach. The stability of the samples during the spectroscopic measurements was ensured by monitoring the absorption spectrum of the Ag SPR (Supplementary Figure 10).”

“The decay kinetics of Ag/ TiO_2 photoholes exhibits primarily a monoexponential behaviour, but we observed a small increase in the signal amplitude at 460 nm in the 10^{-5} – 10^{-2} s timescale (Figure 4A).”

“These values have been obtained through a simulation using the equivalent electric circuit that can be observed inside the Figure 4D. Fitting result obtained at different bias potentials and under dark and illumination conditions are presented in supplementary Figure 5 and Supplementary Table 4. These results are in good agreement with the previously observed through TAS measures, since they confirm a great improvement in charge transfer in the Ag- TiO_2 sample compared with bare TiO_2 .”

“Therefore, the charge separation induced by the electron transfer from the TiO_2 to the metal nanoparticles is sufficient to allow water to be the electron source for the CO_2 reduction, being water the electron donor.”

Regarding to Figure 5

“One of the most interesting properties of Ag/ TiO_2 interfaces is their ability to catalyse photoredox reaction under visible light. However, the optoelectronic pathways involved in this reaction are still under debate in the literature. A critical factor is the photoinduced charges dynamics and their effect on the reactivity and product distribution of CO_2 photoreduction (Figure 5). This influence is clearly observed in the reaction performed using an Ag/ TiO_2 catalyst under visible irradiation ($\lambda \geq 450$ nm), which led to the production of CH_3OH , CH_4 and CO , while bare TiO_2 did not show any photocatalytic activity (Figure 5A).

“Thus, visible light excitation of Ag/ TiO_2 results in shorter-lived transient absorption decay that had smaller signal amplitude compared to that obtained when exciting the sample with UV light. This decay is assigned to a smaller generation of charge

carriers, which explains the lower efficiency of visible-light-driven CO₂ reduction compared to UV excitation.”

(I'm particularly confused about the basis for which the sub-band gap states are deduced, ostensibly a key finding of this manuscript.

Response:

In a first stage an additional density of states is observed in valence band XPS studies when silver is deposited over TiO₂. In addition, this hypothesis is confirmed by DFT calculation of Ag clusters supported on TiO₂ nanoparticles (Figure 2B), which exhibits the formation of induced interface states (IFS). This kind of sub band gap states was also recently observed obtained by Tan et al [24. Tan, S., Argondizzo, A., Ren, J., Liu, L., Zhao, J. & Petek, H. Plasmonic coupling at a metal/semiconductor interface. Nat. Photonics 11, 806–812 (2017).]

I'm also not sure I understand how faster TA decay means fewer carriers generated.)

The TA signal at 460 nm corresponds to the absorption of photogenerated holes in TiO₂. The signal amplitude is proportional to the concentration of charge carriers. At $t = 10^{-5}$ s, the signal amplitude of holes for Ag/TiO₂ excited with UV light is $\sim 10^{-3}$ a. u., and nearly an order of magnitude smaller when excited with visible light. The smaller signal amplitude observed when visible light excitation, can partly be explained by a faster recombination of charge carriers. The electron-hole recombination in TiO₂ starts before the resolution limit of our instrument (observed from ca. 10 ps in J. Phys. Chem. Lett. 2016, 7(19), 3747-3746). Since the recombination of charge carriers is much faster when exciting with visible light compared to UV light, this leads to a much smaller concentration of long-lived electrons, which are essential for the multi-electron reduction of CO₂.

The authors seem to recognize this, as the SI for this manuscript is enormous, 23 pages in all, and frequently cited in the manuscript. If the authors objective was to extract the key findings and summarize them in a short, letter-style format, I'm afraid the attempt failed. Indeed, the attempt may be impossible, given the breadth and depth of the actual work done and the discussions that are required to draw it all together. A much more compelling article would have merged the manuscript and SI into a more thorough explanation of how all these pieces fit together to tell a coherent, compelling story, perhaps for a high impact journal like ACS Nano.

Response:

In order to improve the discussion in the manuscript we have included the Raman section from SI:

“Raman studies (Supplementary Figure 7) show that the fresh Ag/TiO₂ surface is covered by different carbon species adsorbed from ambient air. The observed bands are assigned to linear and bent adsorbed CO₂, bicarbonate as well as bidentate and monodentate carbonate.³⁹ After reaction, some changes are

observed, mainly related to a significant increase of adsorbed CO_3^{2-} and HCO_3^- species and the appearance of formate (1366 cm^{-1}) as possible reaction intermediate⁴⁰ in agreement with with NAP-XPS studies. The increase in the bands due to bent CO_2 structures may be the consequence of the generation of CO_2^- species coordinated to metal ions. It is worth highlighting too the appearance of a new Raman band at a shift of ca. 910 and 985 cm^{-1} assigned to the $\nu_{\text{O-O}}$ vibrational mode of surface peroxo $\text{Ti}(\text{O}_2)^{41}$ and peroxocarbonate species (CO_4^{2-} and $\text{C}_2\text{O}_6^{2-}$)^{39,42} respectively. This observation suggests that the formation of both peroxidized species arise, on the one hand, from hole capture by CO_3^{2-} and HCO_3^- and, on the other hand, from partial oxidation of water. This could prevent O_2 from being detected as the outcome of H_2O oxidation, rendering the oxygen balance a troublesome task in this reaction.”

I would like to point out that 23 pages of SI are collecting additional figures that help to understand the results and discussion outlined in the manuscript. Taking into account the Nature Communications policies, no more than 6 figures can be included in the main paper. _____

Reviewer #3 (Remarks to the Author):

Comments:

This manuscript reports a wide and complex characterization of Ag-TiO₂ system for the CO₂ photoreduction reaction. This reaction is actually a hot topic in photocatalysis. Up to now, the reported results demonstrate that this a challenging reaction. So, hard investigations about the catalyst mechanism is really necessary. For this reason, the opportunity of this manuscript is clear and deserve to be considered in this journal.

However, due to the high number of techniques used, a more concise and clear discussion is needed for communication paper. Some arguments appears repeated along the text.

Response: Thanks to the reviewer for this comment. We have included within the text, some explanations that summarise the main conclusions extracted from each technique.

As a general comment, I would recommend authors to lighten the discussion in some sections, especially "Photoinduced charge transfer" section is somehow confusing. Authors should check the numbering of Figures in this section.

Response: The manuscript has been thoroughly revised, especially the "Photoinduced charge transfer" section to try to report our findings in a clearer way. Changes in the text are highlighted in yellow in the revised manuscript.

The conclusion of the MS is at the end is not clear. The key point is the different photocatalytic behavior of Ag-TiO₂ under UV and vis conditions leading to different products in each case. Authors should emphasize this point as conclusion.

Response: We appreciate the reviewer comment. We have amended the conclusions according to these suggestions (changes in the text are highlighted in yellow in the revised manuscript), to clarify the distinct behaviour of Ag NPs under UV and visible irradiation.

At the end of the MS, the "Methods" section is quite redundant since some parts are repeated in the Supplementary Information. This part should be placed at SI.

Response: As far as we know, a Methods section should be provided for a Nature Communication manuscript. However, we agree that some parts (e.g. catalyst characterization) can be redundant, but the journal policies states that:

"The Methods section appears in all online original research articles and should contain all elements necessary for interpretation and replication of the results.

Methods should be written as concisely as possible and typically do not exceed 3,000 words but may be longer if necessary.”

However in some cases and taking into account the complexity of some of these techniques; it is hard to give a concise explanation. For this reason we have included this information in the SI. In any case if the editorial board considers that we can include all the information contained in SI to the Methods section or in protocol exchange we will be happy to do so.

In Table 3 at SI, CO evolution rate at the first row is not correct, should be 0.7 micromol/g-h. The actual value corresponds to the CO production (11.1 micromol after 15 h) as it is shown in Table 2.

Response: We thank the reviewer for noticing this mistake in Supplementary Table 3. This value has been accordingly corrected in the revised version of the manuscript.

After these considerations, I would recommend the publication of this MS in Nature Communication

REVIEWERS' COMMENTS:

Reviewer #1 (Remarks to the Author):

Thank you for spending time on deeper understanding of the mechanisms from the heterogeneous catalytic view and reassure NAP-XPS results. I accept the manuscript as is only one thing is needed: Authors please put a mechanistic graph or a cartoon about your idea of the mechanism and put it in the manuscript or SI.

Reviewer #2 (Remarks to the Author):

I have carefully reviewed the responses and revision by Collado et al. I appreciate the thoughtfulness of their responses and the numerous changes they have made. I am reluctant to claim that the manuscript is sufficiently improved to be suitable for publication in Nature Communications, but after rereading the revision several times, I am not opposed to its publication. For me, the fundamental problem from the original manuscript remains: too many of the findings are insufficiently explained, and the many references to the supplemental information do not help. (My word processor counted 32 times the authors refer the reader to the 23 page supplement for critical information.) I agree with the other reviewers that the authors have performed an impressive number of measurements and modeling to reach interesting and important conclusions, but I wish the authors could have described them more thoroughly and compellingly. Perhaps because of space limitations, they often could only outline the measurement they performed and the conclusion they reached without guiding the reader to reach the same conclusion for themselves. For example, I still don't understand how to interpret Fig. 2 or reach the conclusions they draw in the manuscript, nor do I understand how they used TAS and DRS to reach the associated conclusions about charge transfer. I do appreciate the additional text they added for Figs. 4 and 5 and the description of the Raman results, all of which has helped. However, some of the new additions raise new questions, such as the discussions about silver loading and the shifts of the TAS and SPR features, and many of the explanations provided in the response to the reviewers could have helped other readers if placed in the revised manuscript. Most of the rest of the additions provide no new information; instead, they helpfully restate intermediate findings as summaries to guide the reader, and I appreciate that as well. There are many typos and several odd word choices (e.g. "incanting," "sudden formation"), especially in the new text of the revision, but these can be fixed. Unfortunately, the space limitation imposed on the authors by Nature Communications cannot be fixed. For these reasons, I still believe this manuscript should be substantially revised to include more detailed descriptions and explanations of their findings for submission to a high impact ACS or RCS journal that supports longer manuscripts and more figures. In spite of this, I don't feel strongly enough about my reservations to write a review so negative that it would preclude publication in NatComm.

Reviewers' comments:

Reviewer #1 (Remarks to the Author):

Comments:

Thank you for spending time on deeper understanding of the mechanisms from the heterogeneous catalytic view and reassure NAP-XPS results. I accept the manuscript as is only one thing is needed:

Authors please put a mechanistic graph or a cartoon about your idea of the mechanism and put it in the manuscript or SI.

We are glad that reviewer #1 is satisfied with the changes made. Starting from mechanisms proposed in the literature and based on the species we have detected in the in-situ experiments, we have elaborated a proposed reaction pathway which is now included in the SI as Supplementary Figure 17 and explained in the main text.

“Under UV irradiation, Ag/TiO₂ is highly selective towards CH₄ (Figure 6A). This improved reactivity to methane is attributed to the electron scavenging ability of Ag NPs, which results in a lifetime of charge carriers increased by 3 orders of magnitude compared to bare TiO₂. In-situ studies support not only this charge transfer but also the formation of C-H intermediates in the pathway to methane following the carbene mechanism (Supplementary Figure 17). Ag NPs are also indispensable for the visible light-induced photoactivity, predominantly towards CH₃OH.”

Supplementary Figure 17. Proposed reaction pathways based on the carbene mechanism¹⁶ and on data obtained from *in-situ* experiments.

Reviewer #2 (Remarks to the Author):

I have carefully reviewed the responses and revision by Collado et al. I appreciate the thoughtfulness of their responses and the numerous changes they have made. I am reluctant to claim that the manuscript is sufficiently improved to be suitable for publication in Nature Communications, but after rereading the revision several times, I am not opposed to its publication. For me, the fundamental problem from the original manuscript remains: too many of the findings are insufficiently explained, and the many references to the supplemental information do not help. (My word processor counted 32 times the authors refer the reader to the 23 page supplement for critical information.) I agree with the other reviewers that the authors have performed an impressive number of measurements and modeling to reach interesting and important conclusions, but I wish the authors could have described them more thoroughly and compellingly. Perhaps because of space limitations, they often could only outline the measurement they performed and the conclusion they reached without guiding the reader to reach the same conclusion for themselves. For example, I still don't understand how to interpret Fig. 2 or reach the conclusions they draw in the manuscript, nor do I understand how they used TAS and DRS to reach the associated conclusions about charge transfer. I do appreciate the additional text they added for Figs. 4 and 5 and the description of the Raman results, all of which has helped. However, some of the new additions raise new questions, such as the discussions about silver loading and the shifts of the TAS and SPR features, and many of the explanations provided in the response to the reviewers could have helped other readers if placed in the revised manuscript. Most of the rest of the additions provide no new information; instead, they helpfully restate intermediate findings as summaries to guide the reader, and I appreciate that as well. There are many typos and several odd word choices (e.g. "incanting," "sudden formation"), especially in the new text of the revision, but these can be fixed. Unfortunately, the space limitation imposed on the authors by Nature Communications cannot be fixed. For these reasons, I still believe this manuscript should be substantially revised to include more detailed descriptions and explanations of their findings for submission to a high impact ACS or RCS journal that supports longer manuscripts and more figures. In spite of this, I don't feel strongly enough about my reservations to write a review so negative that it would preclude publication in NatComm.

We are grateful to reviewer #2 for the great attention paid to our work. Accordingly, we have tried our best to further clarify the path towards our conclusions. Specifically:

In the section entitled "Electronic band structure of the Ag/TiO₂ heterojunction", we have further explained how we have reached our conclusions by linking every piece of information to the theoretical or experimental results they are extracted from, including the different panels of Figure 2.

“This information is essential to understand the light induced processes, including in the photocatalytic reactions. Silver deposition on the titania surface modifies the interface band structure as confirmed by the formation of surface states near the TiO₂ Fermi level (EF). This can be observed in the valence band XPS spectrum (Figure 2A) and also in the flat band potentials determined by the Mott-Schottky plots obtained through EIS measurements (Figure 2C).²⁵ The valence band spectrum of bare titania shows a broad band due to O 2p that is formed by the characteristic contribution of bonding (at 7.6 eV) and non-bonding orbitals (at 5.6 eV). On the other hand, the Ag/TiO₂ sample shows an increase in surface states in the non-bonding region and changes in the VB edge (\approx 0.2 eV), which that are consistent with an interface interaction between surface atoms which that could be assigned to the formation of Ag-O bonds. This hypothesis is confirmed by the electronic structure obtained by density functional theory (DFT) calculation of Ag clusters supported on TiO₂ nanoparticles. Thus, the density of states (DOS) of a bare TiO₂ cluster shows a band gap ($E_g = 3.2$ eV) where in which valence and conduction bands are mainly formed by O 2p and Ti 3d orbitals, respectively (Supplementary Figure 3). Otherwise, Ag/TiO₂ (Figure 2B) exhibits the formation of induced interface states (IFS)²³ in the band gap region. These surface sub-band gap (S_{Eg}) states can be assigned to charge donation from Ag 5s to O 2p neighbouring atoms and Ti 3d orbitals. These S_{Eg} states have been previously described in this kind of materials²⁸ and are consistent with ultrafast (<10 fs) photoinduced hot electrons in TiO₂, rather than electrons photo-generated in Ag and transferred to the semiconductor. The formation of these IFS states could be a cornerstone to explain photocatalytic activity in the visible spectrum region. Because of the narrow differences between the Ag and TiO₂ work functions, depending on morphological or structural properties,²⁷ different band bending scenarios have been previously described for Ag/TiO₂ interfaces.^{22,24} For the present samples, the flat band potentials and therefore the Fermi level (EF) were calculated from the charge transfer capacitance (C_{SC}) obtained from the EIS studies (Figure 2C). The dependence of C_{SC} on bias potential (V) can be described by the Mott-Schottky equation (Supplementary Note 1, equation 1).”

At the beginning of the section entitled “Photoinduced charge transfer”, we have further explained the results obtained from *in-situ* diffuse reflectance spectroscopy (DRS), both under UV and visible irradiation, and their connection with transient absorption spectroscopy (TAS) measurements. In this section, we have also further explained the results obtained from electrochemical impedance spectroscopy (EIS) shown in Figure 4.

In Page 7

“In-situ charge transfer studies using UV-vis diffuse reflectance spectroscopy (DRS) at 365 nm irradiation (Supplementary Figure 8) show the quick formation of a broad band assigned to the surface plasmon resonance (SPR) of silver.^{44,45} Before irradiation, only an intense absorption band with an onset at 380 nm is observed due to the band-gap transition of TiO₂ (c.a. 3.2 eV). After 1 second of UV irradiation, a visible absorption appears as a broad band covering the range 400 - 700 nm and centered at about 500 nm, which is attributed to the SPR absorption of Ag⁰ nanoparticles.^{44,45} During UV irradiation, the intensity of the plasmon band

increases with a red shift in the peak position (~ 80 nm), indicating a slight increase in the Ag particle size.²² It is also noteworthy that after 5 minutes the plasmonic band did not show any significant change, indicating the high rate of Ag NPs photoreduction (Supplementary Figure 8). A concomitant decrease in the intensity of the UV absorption band is also observed. This behavior is representative of the progressive formation of Ag⁰ due to the flow of photogenerated electrons in the conduction band of TiO₂ towards surface Ag oxides. The formation of Ag⁰ species is consistent with previous NAP-XPS observations (Supplementary Figure 6), and it is clearly noticeable by a darkening of the sample due to Ag NPs photochromism (Figure 3A inset).⁴⁶”

In Page 8

“Photocurrent densities obtained in a three-electrode configuration cell at different bias potentials and under UV illumination exhibit an improvement of ca. 0.1 mA cm⁻² when Ag NPs are supported on TiO₂ (Figure 4C). This confirms that the presence of silver nanoparticles decreases the electron recombination and enhances the charge transfer and separation. Moreover, electrochemical impedance spectroscopy (EIS) has been performed under dark and illumination conditions at different bias potentials, and the Nyquist plots at 0 V vs Ag/AgCl (Figure 4D) show that the resistance of the TiO₂ space charge region decreases from 81 to 36 ± 0.5 kΩ when Ag NPs are supported on bare TiO₂. These values have been obtained through a simulation using the equivalent electric circuit depicted in the inset of Figure 4D. The fitting results obtained at different bias potentials and under dark and illumination conditions are presented in Supplementary Figure 5 and Supplementary Table 4. These results are in good agreement with previous TAS observations, since they confirm a significant improvement in Ag/TiO₂ charge transfer compared to bare TiO₂.”

In addition, we have included all the explanation previously included in SI related to methodology in the main manuscript methods section.